# PRIVACY-PRESERVING LOGISTIC REGRESSION TRAINING WITH A FASTER GRADIENT VARIANT

## ABSTRACT

Training logistic regression over encrypted data has been a compelling approach in addressing security concerns for several years. In this paper, we introduce an efficient gradient variant, called $quadratic\ gradient$, for privacy-preserving logistic regression training. We enhance Nesterov's Accelerated Gradient (NAG), Adaptive Gradient Algorithm (Adagrad) and Adam algorithms by incorporating their quadratic gradients and evaluate these improved algorithms on various datasets. Experimental results demonstrate that the enhanced algorithms achieve significantly improved convergence speed compared to traditional first-order gradient methods. Moreover, we applied the enhanced NAG method to implement homomorphic logistic regression training, achieving comparable results within just 4 iterations. There is a good chance that the quadratic gradient approach could integrate first-order gradient descent/ascent algorithms with the second-order Newton-Raphson methods, and that it could be applied to a wide range of numerical optimization problems.

## 1 INTRODUCTION

Given a person's healthcare data related to a certain disease, we can train a logistic regression (LR) model capable of telling whether or not this person is likely to develop this disease. However, such personal health information is highly private to individuals. The privacy concern, therefore, becomes a major obstacle for individuals to share their biomedical data. The most secure solution is to encrypt the data into ciphertexts first by Homomorphic Encryption (HE) and then securely outsource the ciphertexts to the cloud, without allowing the cloud to access the data directly. iDASH is an annual competition that aims to call for implementing interesting cryptographic schemes in a biological context. Since 2014, iDASH has included the theme of genomics and biomedical privacy. The third track of the 2017 iDASH competition and the second track of the 2018 iDASH competition were both to develop HE-based solutions for building an LR model over encrypted data.

Several studies on logistic regression models are based on homomorphic encryption. Kim et al. (2018b) discussed the problem of performing LR training in an encrypted environment. They employed full-batch gradient descent during the training process and utilized the least-squares method to approximate the sigmoid function. In the 2017 iDASH competition, Bonte & Vercauteren (2018), Kim et al. (2018a), Chen et al. (2018), and Crawford et al. (2018) all addressed the same problem explored by Kim et al. (2018b). In the 2018 iDASH competition, Kim et al. (2019b) and Blatt et al. (2019) further developed the problem, focusing on efficient packing and a semi-parallel algorithm. There are other related works (Kim et al., 2019a; Bergamaschi et al., 2019; Ogilvie et al., 2020) focusing on various aspects but the papers most relevant to this work are (Bonte & Vercauteren, 2018) and (Kim et al., 2018a). Bonte & Vercauteren (2018) developed a practical algorithm called the simplified fixed Hessian (SFH) method. Our study extends their work and adopts the ciphertext packing technique proposed by Kim et al. (2018a) for efficient homomorphic computation.

Our specific contributions in this paper are as follows:

1. We propose a new gradient variant, quadratic gradient, which can combine the first-order gradient algorithms and the second-order Newton-Raphson method (aka Newton's method).

2. We develop three enhanced gradient algorithms by equipping the original ones with quadratic gradient. The resulting algorithms all converge and, in most cases, demonstrate strong performance in terms of convergence speed.

3. We implement privacy-preserving logistic regression training using the enhanced NAG method, to our best knowledge, which seems to be a great choice without compromising much on computation and storage.

## 2 PRELIMINARIES

We use the square brackets "[ ]" to denote the index of a vector or matrix element in what follows. For example, for a vector $\boldsymbol{v} \in \mathbb{R}^{(n)}$ and a matrix $M \in \mathbb{R}^{m \times n}$, $\boldsymbol{v}[i]$ or $\boldsymbol{v}_{[i]}$ means the $i$-th element of vector $\boldsymbol{v}$ and $M[i][j]$ or $M_{[i][j]}$ the $j$-th element in the $i$-th row of $M$.

### 2.1 FULLY HOMOMORPHIC ENCRYPTION

Fully Homomorphic Encryption (FHE) is a type of cryptographic scheme that can be used to compute an arbitrary number of additions and multiplications directly on the encrypted data. It was not until 2009 that Gentry constructed the first FHE scheme via a bootstrapping operation (Gentry, 2009). FHE schemes themselves are computationally time-consuming; the choice of dataset encoding matters likewise to the efficiency. In addition to these two limits, how to manage the magnitude of plaintext (Jäschke & Armknecht, 2016) also contributes to the slowdown. Cheon et al. (2017) proposed a method to construct an HE scheme with a `rescaling` procedure which could eliminate this technical bottleneck effectively. We adopt their open-source implementation `HEAAN` while implementing our homomorphic LR algorithms. In addition, it is inevitable to pack a vector of multiple plaintexts into a single ciphertext for yielding a better amortized time of homomorphic computation. `HEAAN` supports a parallel technique (aka `SIMD`) to pack multiple complex numbers in a single polynomial and provides rotation operation on plaintext slots. The underlying HE scheme in `HEAAN` is well described in (Kim et al., 2018a;b; Han et al., 2019).

### 2.2 DATABASE ENCODING METHOD

Kim et al. (2018a) proposed an efficient and promising database-encoding method by using `SIMD` technique, which could make full use of the computation and storage resources. Suppose a database contains a training dataset $Z$ comprising $n$ samples and $(1 + d)$ covariates. The dataset matrix $Z$ is packed into a single ciphertext in a row-by-row manner.

When employing this encoding scheme, we can manipulate the data matrix $Z$ through HE operations on the ciphertext $Enc[Z]$, utilizing only three HE operations - rotation, addition, and multiplication. For instance, if we wish to isolate the first column of $Enc[Z]$ and exclude the other columns, we can create a constant matrix $F$ with ones in the first column and zeros elsewhere. Multiplying $Enc[Z]$ by $F$ will yield the desired ciphertext.

Han et al. (2019) mentioned several basic but important operations used by Kim et al. (2018a) in their implementation, such as a procedure named "`SumColVec`" to compute the summation of the columns of a matrix. With these fundamental operations, more intricate computations, like computing gradients in logistic regression models, become achievable.

### 2.3 LOGISTIC REGRESSION MODEL

Logistic regression is widely used in binary classification tasks to infer whether a binary-valued variable belongs to a certain class or not. LR can be generalized from linear regression (Murphy, 2012) by mapping the whole real line ($\boldsymbol{\beta}^\top \mathbf{x}$) to $(0, 1)$ via the sigmoid function $\sigma(z) = 1/(1 + \exp(-z))$, where the vector $\boldsymbol{\beta} \in \mathbb{R}^{(1+d)}$ is the main parameter of LR and the vector $\mathbf{x} = (1, x_1, \ldots, x_d) \in \mathbb{R}^{(1+d)}$ the input covariate. Thus logistic regression can be formulated with the class label $y \in \{\pm 1\}$ as follows:

$$\Pr(y = +1|\mathbf{x}, \boldsymbol{\beta}) = \sigma(\boldsymbol{\beta}^\top \mathbf{x}) \qquad = \frac{1}{1 + e^{-\boldsymbol{\beta}^\top \mathbf{x}}},$$

$$\Pr(y = -1|\mathbf{x}, \boldsymbol{\beta}) = 1 - \sigma(\boldsymbol{\beta}^\top \mathbf{x}) \quad = \frac{1}{1 + e^{+\boldsymbol{\beta}^\top \mathbf{x}}}.$$

LR sets a threshold (usually $0.5$) and compares its output with it to decide the resulting class label.

The logistic regression problem can be transformed into an optimization problem that seeks a parameter $\boldsymbol{\beta}$ to maximize $L(\boldsymbol{\beta}) = \prod_{i=1}^n \Pr(y_i|\mathbf{x}_i, \boldsymbol{\beta})$ or its log-likelihood function $l(\boldsymbol{\beta})$ for convenience in the calculation:

$$l(\boldsymbol{\beta}) = \ln L(\boldsymbol{\beta}) = -\sum_{i=1}^n \ln(1 + e^{-y_i \boldsymbol{\beta}^\top \mathbf{x}_i}),$$

where $n$ is the number of examples in the training dataset. LR does not have a closed form of maximizing $l(\boldsymbol{\beta})$ and two main methods are adopted to estimate the parameters of an LR model: (a) gradient descent method via the gradient; and (b) Newton's method by the Hessian matrix. The gradient and Hessian of the log-likelihood function $l(\boldsymbol{\beta})$ are given by, respectively:

$$\nabla_{\boldsymbol{\beta}} l(\boldsymbol{\beta}) = \sum_i (1 - \sigma(y_i \boldsymbol{\beta}^\top \mathbf{x}_i)) y_i \mathbf{x}_i,$$

$$\nabla_{\boldsymbol{\beta}}^2 l(\boldsymbol{\beta}) = \sum_i (y_i \mathbf{x}_i)(\sigma(y_i \boldsymbol{\beta}^\top \mathbf{x}_i) - 1) \sigma(y_i \boldsymbol{\beta}^\top \mathbf{x}_i)(y_i \mathbf{x}_i)$$

$$= X^\top S X,$$

where $S$ is a diagonal matrix with entries $S_{ii} = (\sigma(y_i \boldsymbol{\beta}^\top \mathbf{x}_i) - 1)\sigma(y_i \boldsymbol{\beta}^\top \mathbf{x}_i)$ and $X$ the dataset.

The log-likelihood function $l(\boldsymbol{\beta})$ of LR has at most a unique global maximum (Allison, 2008), where its gradient is zero. Newton's method is a second-order technique to numerically find the roots of a real-valued differentiable function, and thus can be used to solve the $\boldsymbol{\beta}$ in $\nabla_{\boldsymbol{\beta}} l(\boldsymbol{\beta}) = 0$ for LR.

## 3 TECHNICAL DETAILS

It is quite time-consuming to compute the Hessian matrix and its inverse in Newton's method for each iteration. One way to limit this downside is to replace the varying Hessian with a fixed matrix $\bar{H}$. This novel technique is called the fixed Hessian Newton's method. Böhning & Lindsay (1988) have shown that the convergence of Newton's method is guaranteed as long as $\bar{H} \leq \nabla_{\boldsymbol{\beta}}^2 l(\boldsymbol{\beta})$, where $\bar{H}$ is a symmetric negative-definite matrix independent of $\boldsymbol{\beta}$ and "$\leq$" denotes the Loewner ordering in the sense that the difference $\nabla_{\boldsymbol{\beta}}^2 l(\boldsymbol{\beta}) - \bar{H}$ is non-negative definite. With such a fixed Hessian matrix $\bar{H}$, the iteration for Newton's method can be simplified to:

$$\boldsymbol{\beta}_{t+1} = \boldsymbol{\beta}_t - \bar{H}^{-1} \nabla_{\boldsymbol{\beta}} l(\boldsymbol{\beta}).$$

Böhning and Lindsay also suggest the fixed matrix $\bar{H} = -\frac{1}{4} X^\top X$ is a good lower bound for the Hessian of the log-likelihood function $l(\boldsymbol{\beta})$ in LR.

### 3.1 SIMPLIFIED FIXED HESSIAN

Bonte & Vercauteren (2018) simplify this lower bound $\bar{H}$ further due to the need for inverting the fixed Hessian in the encrypted domain. They replace the matrix $\bar{H}$ with a diagonal matrix $B$ whose diagonal elements are simply the sums of each row in $\bar{H}$. They also suggest a specific order of calculation to optimize the computation of $B$ more efficiently. Their new approximation $B$ of the fixed Hessian is:

$$B = \begin{bmatrix} \sum_{i=0}^d \bar{h}_{0i} & 0 & \cdots & 0 \\ 0 & \sum_{i=0}^d \bar{h}_{1i} & \cdots & 0 \\ \vdots & \vdots & \ddots & \vdots \\ 0 & 0 & \cdots & \sum_{i=0}^d \bar{h}_{di} \end{bmatrix},$$

where $\bar{h}_{ki}$ is the element of $\bar{H}$. This diagonal matrix $B$ is in a very simple form and can be obtained from $\bar{H}$ without much difficulty. The inverse of $B$ can even be approximated in the encrypted form by computing the inverse of each diagonal element of $B$ using an iterative Newton's method with an appropriate initial value. Their simplified fixed Hessian method can be formulated as follows:

$$\boldsymbol{\beta}_{t+1} = \boldsymbol{\beta}_t - B^{-1} \cdot \nabla_{\boldsymbol{\beta}} l(\boldsymbol{\beta}),$$

$$= \boldsymbol{\beta}_t - \begin{bmatrix} b_{00} & 0 & \ldots & 0 \\ 0 & b_{11} & \ldots & 0 \\ \vdots & \vdots & \ddots & \vdots \\ 0 & 0 & \ldots & b_{dd} \end{bmatrix} \cdot \begin{bmatrix} \nabla_0 \\ \nabla_1 \\ \vdots \\ \nabla_d \end{bmatrix} = \boldsymbol{\beta}_t - \begin{bmatrix} b_{00} \cdot \nabla_0 \\ b_{11} \cdot \nabla_1 \\ \vdots \\ b_{dd} \cdot \nabla_d \end{bmatrix},$$

where $b_{ii}$ is the reciprocal of $\sum_{i=0}^{d} \bar{h}_{0i}$ and $\nabla_i$ is the element of $\nabla_{\boldsymbol{\beta}} l(\boldsymbol{\beta})$.

Consider a special situation: if all the elements $b_{00}, \ldots, b_{dd}$ had the same value $-\eta$ with $\eta > 0$, the iterative formula of the SFH method could be given as:

$$\boldsymbol{\beta}_{t+1} = \boldsymbol{\beta}_t - (-\eta) \cdot \begin{bmatrix} \nabla_0 \\ \nabla_1 \\ \vdots \\ \nabla_d \end{bmatrix} = \boldsymbol{\beta}_t + \eta \cdot \nabla_{\boldsymbol{\beta}} l(\boldsymbol{\beta}),$$

which is the same as the formula of the naive gradient *ascent* method. Such a coincidence not only helps generate the idea behind this work but also leads us to believe that there is a connection between the Hessian matrix and the learning rate of the gradient (descent) method.

We regard $B^{-1} \cdot \nabla_i$ as a novel enhanced gradient variant and allocate a distinct learning rate to it. As long as we ensure that this new learning rate decreases from a positive floating-point number greater than 1 (such as 2) to 1 in a bounded number of iteration steps, the fixed Hessian Newton's method guarantees the algorithm will converge eventually.

The SFH method proposed by Bonte & Vercauteren (2018) has two limitations: (a) in the construction of the simplified fixed Hessian, all entries in the symmetric matrix $\bar{H}$ need to be non-positive. For machine learning applications, datasets are typically normalized in advance to the range [0,1], satisfying the convergence condition of the SFH method. However, in other cases, such as numerical optimization, this condition may not always hold; and (b) the simplified fixed Hessian matrix $B$, as well as the fixed Hessian $\bar{H} = -\frac{1}{4} X^\top X$, can still be singular, especially when the dataset is a high-dimensional sparse matrix, such as the MNIST datasets. We extend SFH by removing these limitations to generalize this simplified fixed Hessian to be invertible in any case and propose a faster gradient variant, which we term *quadratic gradient*.

## 3.2 QUADRATIC GRADIENT DEFINITION

Suppose that a differentiable scalar-valued function $F(\mathbf{x})$ has its gradient $\boldsymbol{g}$ and Hessian matrix $H$, with any matrix $\bar{H} \leq H$ in the Loewner ordering for a *maximization* problem as follows:

$$\boldsymbol{g} = \begin{bmatrix} g_0 \\ g_1 \\ \vdots \\ g_d \end{bmatrix}, \quad H = \begin{bmatrix} \nabla_{00}^2 & \nabla_{01}^2 & \cdots & \nabla_{0d}^2 \\ \nabla_{10}^2 & \nabla_{11}^2 & \cdots & \nabla_{1d}^2 \\ \vdots & \vdots & \ddots & \vdots \\ \nabla_{d0}^2 & \nabla_{d1}^2 & \cdots & \nabla_{dd}^2 \end{bmatrix}, \quad \bar{H} = \begin{bmatrix} \bar{h}_{00} & \bar{h}_{01} & \ldots & \bar{h}_{0d} \\ \bar{h}_{10} & \bar{h}_{11} & \ldots & \bar{h}_{1d} \\ \vdots & \vdots & \ddots & \vdots \\ \bar{h}_{d0} & \bar{h}_{d1} & \ldots & \bar{h}_{dd} \end{bmatrix},$$

where $\nabla_{ij}^2 = \nabla_{ji}^2 = \frac{\partial^2 F}{\partial x_i \partial x_j}$. We construct a new diagnoal Hessian matrix $\tilde{B}$ with each diagnoal element $\tilde{B}_{kk}$ being $-\epsilon - \sum_{i=0}^{d} |\bar{h}_{ki}|$,

$$\tilde{B} = \begin{bmatrix} -\epsilon - \sum_{i=0}^{d} |\bar{h}_{0i}| & 0 & \ldots \\ 0 & -\epsilon - \sum_{i=0}^{d} |\bar{h}_{1i}| & \ldots \\ \vdots & \vdots & \ddots \\ 0 & 0 & \ldots \end{bmatrix},$$

where $\epsilon$ is a small positive constant to avoid division by zero (usually set to $1e - 8$).

As long as $\tilde{B}$ satisfies the convergence condition of the aforementioned fixed Hessian method, namely $\tilde{B} \preceq H$, we can use this approximation $\tilde{B}$ of the Hessian matrix as a lower bound. Given that we have already assumed $\bar{H} \preceq H$, it suffices to demonstrate that $\tilde{B} \preceq \bar{H}$. We establish $\tilde{B} \preceq \bar{H}$ in a manner similar to that employed by Bonte & Vercauteren (2018).

**Lemma 3.1.** *Let $A \in \mathbb{R}^{n \times n}$ be a symmetric matrix, and let $B$ be the diagonal matrix whose diagonal entries $B_{kk} = -\epsilon - \sum_i |A_{ki}|$ for $k = 1, \ldots, n$, then $B \preceq A$.*

*Proof.* By definition of the Loewner ordering, we have to prove the difference matrix $C = A - B$ is non-negative definite, which means that all the eigenvalues of $C$ need to be non-negative. By construction of $C$ we have that $C_{ij} = A_{ij} + \epsilon + \sum_{k=1}^{n} |A_{ik}|$ for $i = j$ and $C_{ij} = A_{ij}$ for $i \neq j$. By means of Gerschgorin's circle theorem, we can bound every eigenvalue $\lambda$ of $C$ in the sense that $|\lambda - C_{ii}| \leq \sum_{i \neq j} |C_{ij}|$ for some index $i \in \{1, 2, \ldots, n\}$. We conclude that $\lambda \geq A_{ii} + \epsilon + |A_{ii}| \geq \epsilon > 0$ for all eigenvalues $\lambda$ and thus that $B \preceq A$. $\square$

**Definition 3.2** (Quadratic Gradient). Given the previously defined $\tilde{B}$, we define the quadratic gradient as $G = \bar{B} \cdot g$ with a new learning rate $N_t$, where $\bar{B}$ is a diagonal matrix with diagonal entries $\bar{B}_{kk} = 1/|\tilde{B}_{kk}|$. The learning rate $N_t$ should always be no less than 1 and is designed to decrease to 1 over a limited number of iterations. It is important to note that $G$ remains a column vector of the same dimension as the gradient $g$. To maximize the function $F(\mathbf{x})$, we can employ the iterative formula: $\mathbf{x}_{t+1} = \mathbf{x}_t + N_t \cdot G$, similar to the naive gradient approach.

Minimizing the function $F(x)$ is equivalent to maximizing the function $-F(x)$. In this context, we need to construct $\tilde{B}$ using either a good *lower* bound $\bar{H}$ for the Hessian $-H$ of $-F(x)$ or a good *upper* bound $\bar{H}$ for the Hessian $H$ of $F(x)$. It is worth noting that $\bar{H}$ can be taken as the Hessian $H$ itself, thereby eliminating the need to find either lower or upper bounds.

Böhning & Lindsay (1988) did not propose a systematic method for determining or constructing a constant Hessian approximation for their fixed Hessian method. This may be due to the absence of such matrices for most objective functions. We note that our quadratic gradient approach does not require a constant (fixed) Hessian replacement and instead allows for the use of the original Hessian itself to construct our varying diagonal matrix $\tilde{B}$ for the quadratic gradient. Under this less restrictive condition, we present a methodical approach to evaluate whether a fixed Hessian matrix can be employed for efficient computation. Specifically, we first construct $\tilde{B}$ from the Hessian of the given objective function and then check if, for each diagonal element $\tilde{B}_{kk}$, there is a constant maximum for $\bar{B}_{kk} = 1/|\tilde{B}_{kk}|$ (or a constant minimum $|\tilde{B}_{kk}|$). If each diagonal element $|\tilde{B}_{kk}|$ achieves its minimum $|\tilde{M}_{kk}|$, then a constant diagonal matrix with the corresponding diagonal elements $|\tilde{M}_{kk}|$ would satisfy the convergence condition for the SFH method. Finally, we should evaluate the performance of such a fixed Hessian replacement, as it might not always provide a good bound. This search method can be applied to any optimization function to identify a fixed Hessian. However, the absence of a constant Hessian replacement via this method does not necessarily imply that such an approximation does not exist.

### 3.3 QUADRATIC GRADIENT ALGORITHMS

Quadratic gradient can be used to enhance various first-order gradient algorithms:

1. NAG is a different variant of the momentum method to give the momentum term much more prescience. The iterative formulas of the gradient *ascent* method for NAG are as follows:

$$V_{t+1} = \boldsymbol{\beta}_t + \eta_t \cdot \nabla J(\boldsymbol{\beta}_t), \tag{3}$$

$$\boldsymbol{\beta}_{t+1} = (1 - \gamma_t) \cdot V_{t+1} + \gamma_t \cdot V_t, \tag{4}$$

where $V_{t+1}$ is the intermediate variable used for updating the final weight $\boldsymbol{\beta}_{t+1}$ and $\gamma_t \in (0, 1)$ is a smoothing parameter of moving average to evaluate the gradient at an approximate future position (Kim et al., 2018a). The enhanced NAG is to replace equation 3 with $V_{t+1} = \boldsymbol{\beta}_t + N_t \cdot G$, where $N_t$ is the new learning rate for the enhanced algorithm usually setted to $1 + \eta_t$. Our enhanced NAG is described in Algorithm 1.

---

**Algorithm 1** The Enhanced Nesterov's Accelerated Gradient Algorithm

---

**Input:** training dataset $X \in \mathbb{R}^{n \times (1+d)}$; training label $Y \in \mathbb{R}^{n \times 1}$; learning rate $lr \in \mathbb{R}$; and the number $\kappa$ of iterations;

**Output:** the parameter vector $V \in \mathbb{R}^{(1+d)}$

1: Set $\bar{H} \leftarrow -\frac{1}{4}X^\top X$, $V \leftarrow \mathbf{0}$, $W \leftarrow \mathbf{0}$, $\bar{B} \leftarrow \mathbf{0}$

2: **for** $i := 0$ to $d$ **do**                         $\triangleright$ $\epsilon$ is a small positive constant such as $1e-8$

3:    $\bar{B}[i][i] \leftarrow \epsilon$

4:    **for** $j := 0$ to $d$ **do**

5:        $\bar{B}[i][i] \leftarrow \bar{B}[i][i] + |\bar{H}[i][j]|$

6:    **end for**

7: **end for**

8: Set $alpha_0 \leftarrow 0.01$, $alpha_1 \leftarrow 0.5 \times (1 + \sqrt{1 + 4 \times alpha_0^2})$

9: **for** $count := 1$ to $\kappa$ **do**

10:    Set $Z \leftarrow \mathbf{0}$                      $\triangleright$ $Z \in \mathbb{R}^n$ will store the inputs for Sigmoid function

11:    **for** $i := 1$ to $n$ **do**

12:        **for** $j := 0$ to $d$ **do**

13:            $Z[i] \leftarrow Z[i] + Y[i] \times V[j] \times X[i][j]$

14:        **end for**

15:    **end for**

16:    Set $\boldsymbol{\sigma} \leftarrow \mathbf{0}$                    $\triangleright$ $\boldsymbol{\sigma} \in \mathbb{R}^n$ will store the outputs of Sigmoid function

17:    **for** $i := 1$ to $n$ **do**

18:        $\boldsymbol{\sigma}[i] \leftarrow 1/(1 + \exp(-Z[i]))$

19:    **end for**

20:    Set $\boldsymbol{g} \leftarrow \mathbf{0}$

21:    **for** $j := 0$ to $d$ **do**

22:        **for** $i := 1$ to $n$ **do**

23:            $\boldsymbol{g}[j] \leftarrow \boldsymbol{g}[j] + (1 - \boldsymbol{\sigma}[i]) \times Y[i] \times X[i][j]$

24:        **end for**

25:    **end for**

26:    Set $G \leftarrow \mathbf{0}$

27:    **for** $j := 0$ to $d$ **do**

28:        $G[j] \leftarrow \bar{B}[j][j] \times \boldsymbol{g}[j]$

29:    **end for**

30:    Set $\eta \leftarrow (1 - alpha_0)/alpha_1$, $\gamma \leftarrow (\frac{10}{1+t})/n$

31:    **for** $j := 0$ to $d$ **do**

32:        $w_{temp} \leftarrow V[j] + (1 + \gamma) \times G[j]$, $V[j] \leftarrow (1 - \eta) \times w_{temp} + \eta \times W[j]$, $W[j] \leftarrow w_{temp}$

33:    **end for**

34:    $alpha_0 \leftarrow alpha_1$, $alpha_1 \leftarrow 0.5 \times (1 + \sqrt{1 + 4 \times alpha_0^2})$

35: **end for**

36: Return $V$

---

2. Adagrad is a gradient-based algorithm suitable for dealing with sparse data. The updated operations of Adagrad and its quadratic-gradient version, for every parameter $\beta_{[i]}$ at each iteration step $t$, are as follows, respectively:

$$\boldsymbol{\beta}_{[i]}^{(t+1)} = \boldsymbol{\beta}_{[i]}^{(t)} - \frac{\eta_t}{\epsilon + \sqrt{\sum_{k=1}^{t} \boldsymbol{g}_{[i]}^{(t)} \cdot \boldsymbol{g}_{[i]}^{(t)}}} \cdot \boldsymbol{g}_{[i]}^{(t)},$$

$$\boldsymbol{\beta}_{[i]}^{(t+1)} = \boldsymbol{\beta}_{[i]}^{(t)} - \frac{N_t}{\epsilon + \sqrt{\sum_{k=1}^{t} G_{[i]}^{(t)} \cdot G_{[i]}^{(t)}}} \cdot G_{[i]}^{(t)}.$$

3. Adam is an optimization algorithm that combines the benefits of Adagrad, which adapts learning rates for each parameter, with momentum, allowing it to converge faster and handle sparse gradients more effectively. Similar to how Adagrad transitions to its enhanced version, the update formula for enhanced Adam, also employs a new learning rate $N_t$ and replaces the gradient with its quadratic form.

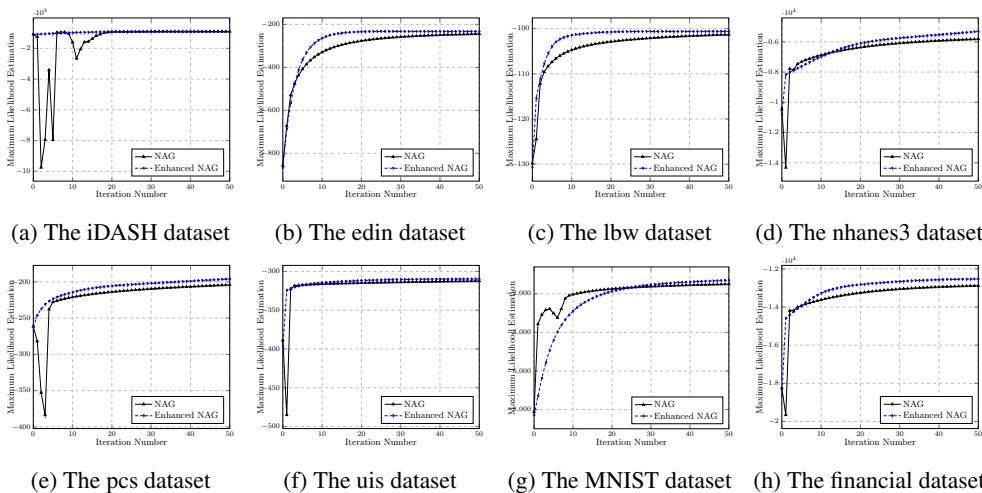

Figure 1: The training results of NAG and Enhanced NAG in the clear.

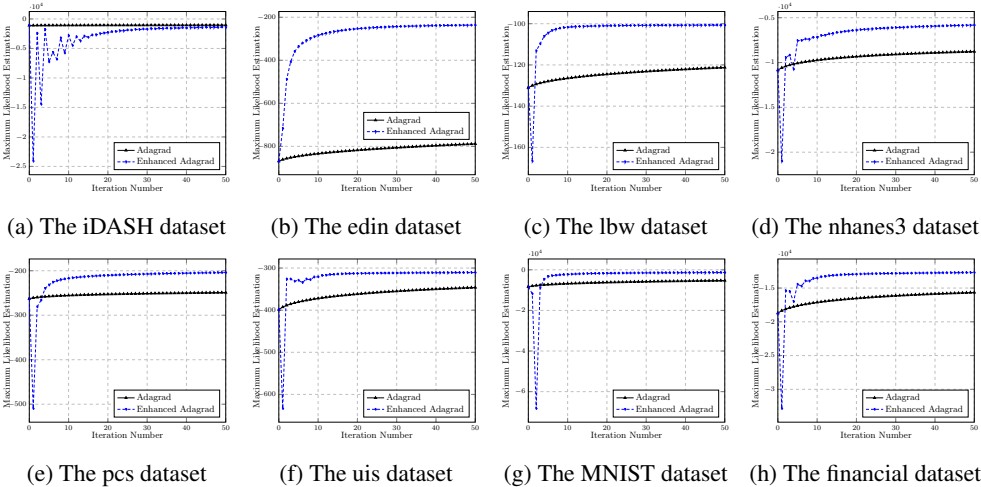

Figure 2: The training results of Adagrad and Enhanced Adagrad in the clear.

**Performance Evaluation** We evaluate the performance of various algorithms in a non-encrypted context using Python on a desktop computer with an Intel Core G640 CPU at 1.60 GHz and 7.3 GB of RAM. Given that our focus is on the convergence speed of the algorithms during training, we use the loss function $l(\boldsymbol{\beta})$, specifically maximum likelihood estimation (MLE), as the only indicator. We evaluate six algorithms—NAG, Adagrad, Adam, and their quadratic-gradient variants (denoted as Enhanced NAG, Enhanced Adagrad, and Enhanced Adam, respectively)—using the datasets adopted by Kim et al. (2018a): the iDASH genomic dataset (iDASH), the Myocardial Infarction dataset from Edinburgh (edin), the Low Birth Weight Study (lbw), Nhanes III (nhanes3), the Prostate Cancer Study (pcs), and the Umaru Impact Study datasets (uis). The genomic dataset from the third task of the 2017 iDASH competition comprises 1,579 records, each featuring 103 binary genotypes and a binary phenotype indicating whether the patient has cancer. The remaining five datasets each contain a single binary dependent variable. We also evaluate these algorithms on two large datasets from Han et al. (2019): a real financial dataset with 422,108 samples and 200 features, and the restructured public MNIST dataset, which includes 11,982 samples from the training set with 196 features. For a fair comparison with the baseline work by Kim et al. (2018a), the enhanced NAG algorithm employs the learning rate $1 + \frac{10}{1+t}$, consistent with Kim et al.'s choice of $\frac{10}{1+t}$ for their learning rate. The enhanced Adagrad algorithm utilizes $N_t = 1 + 0.01$ as the learning rate, while the enhanced Adam algorithm adopts the following learning rate settings: $\alpha = 1 + 0.001$,

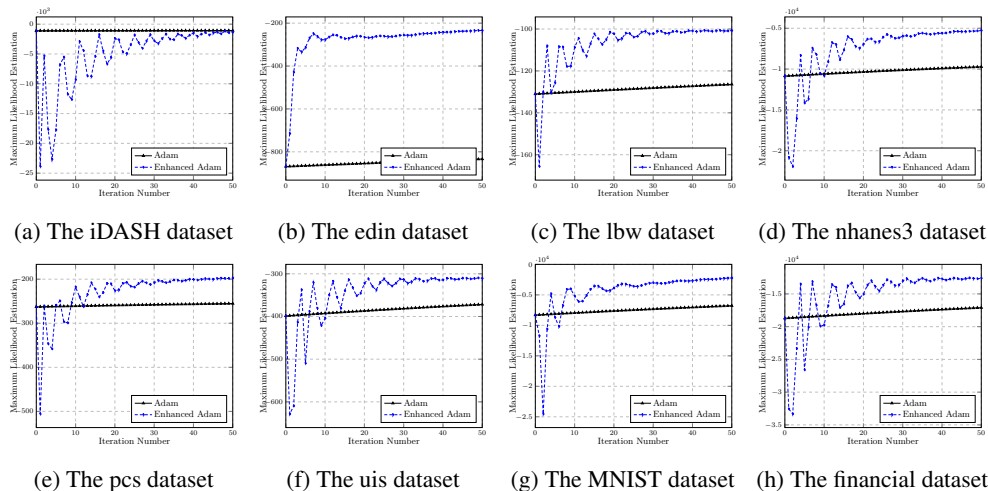

(a) The iDASH dataset    (b) The edin dataset    (c) The lbw dataset    (d) The nhanes3 dataset

(e) The pcs dataset    (f) The uis dataset    (g) The MNIST dataset    (h) The financial dataset

Figure 3: The training results of Adam and Enhanced Adam in the clear.

$\beta_1 = 0.9$, and $\beta_2 = 0.999$. Refer to (Kingma & Ba, 2014) for further details regarding these parameters. In our experiments, we consistently use $\bar{H} = -\frac{1}{4}X^\top X$ to construct $\tilde{B}$ for the function $l(\boldsymbol{\beta})$.

From the empirical results presented in Figures 1, 2, and 3, we conclude that all the enhanced algorithms converge, and that the enhanced NAG algorithm consistently outperforms the original NAG method in terms of convergence rate. Although the enhanced Adagrad and Adam methods do not achieve faster convergence than their original counterparts on the iDASH genomic dataset, they demonstrate clear advantages in all other cases. The reason for this discrepancy is beyond the scope of this paper and will be addressed as part of future work. One possible explanation for the superior performance of the enhanced algorithms is that the quadratic gradient integrates curvature information into first-order gradient methods.

An important observation is that the enhanced algorithms demonstrate better performance with learning rates between 1 and 2 compared to other values. When the learning rate exceeds 3, the algorithm is nearly guaranteed not to converge. For quadratic gradient algorithms, we recommend employing an exponentially decaying learning rate, such as $1 + A \cdot \gamma^t$, where $t$ denotes the iteration number, $A$ is a positive constant typically set to no less than 1, and $\gamma$ is a positive number less than 1 that controls the rate of decay.

**Gradient And Quadratic Gradient** We executed the raw gradient ascent algorithm and the raw quadratic gradient algorithm on the lbw dataset using various learning rates. Figure 4 presents the detailed results of this experiment. It was precisely these results that directly inspired the authors to develop the concept of the quadratic gradient. Similar to the gradient, the quadratic gradient, which incorporates curvature, also demonstrates smooth progression. In particular, as the learning rate is gradually increased, both the gradient and the quadratic gradient reveal corresponding gradual changes in performance, rather than sudden jumps.

## 4   SECURE TRAINING

We adopt the enhanced NAG method to implement secure logistic regression training based on HE. The difficulty in applying the quadratic gradient is to invert the diagonal matrix $\tilde{B}$ in order to obtain $\bar{B}$. We leave the computation of matrix $\bar{B}$ to data owner and let the data owner upload the ciphertext encrypting the $\bar{B}$ to the cloud. Since data owner has to prepare the dataset and normalize it, it would also be practicable for the data owner to calculate the $\bar{B}$ owing to no leaking of sensitive data information.

Privacy-preserving logistic regression training based on HE techniques faces a difficult dilemma that no homomorphic schemes are capable of directly calculating the sigmoid function in the LR

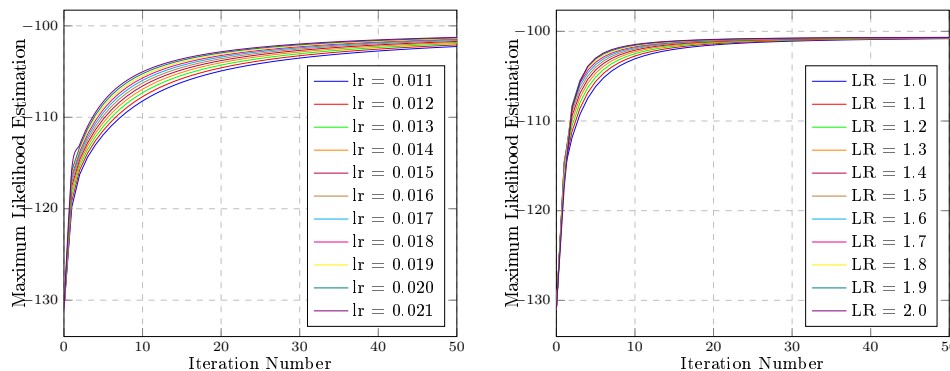

(a) The raw first-order gradient ascent algorithm with iterative formulas: $\boldsymbol{\beta}_{t+1} = \boldsymbol{\beta}_t + \mathrm{lr} \cdot \boldsymbol{g}$ (b) The raw quadratic gradient ascent algorithm with iterative formulas: $\boldsymbol{\beta}_{t+1} = \boldsymbol{\beta}_t + \mathrm{LR} \cdot G$

Figure 4: The training outcomes for both the raw gradient algorithm and the raw quadratic gradient ascent algorithm in the clear setting, conducted on the lbw dataset.

model. A common solution is to replace the sigmoid function with a polynomial approximation by using the widely adopted least-squares method. We can call a function named " `polyfit(·)` " in the Python package Numpy to fit the polynomial in a least-square sense. We adopt the degree 5 polynomial approximation $g(x)$ developed by Kim et al. (2018a), utilizing the least squares approach to approximate the sigmoid function over the interval $[-8, 8]$: $g(x) = 0.5 + 0.19131 \cdot x - 0.0045963 \cdot x^3 + 0.0000412332 \cdot x^5$ .

Given the training dataset $\mathrm{X} \in \mathbb{R}^{n \times (1+d)}$ and training label $\mathrm{Y} \in \mathbb{R}^{n \times 1}$, we adopt the same method that Kim et al. (2018a) used to encrypt the data matrix consisting of the training data combined with training-label information into a single ciphertext $\mathrm{ct}_Z$. The weight vector $\beta^{(0)}$ consisting of zeros and the diagnoal elements of $\bar{B}$ are copied $n$ times to form two matrices. The data owner then encrypt the two matrices into two ciphertexts $\mathrm{ct}_\beta^{(0)}$ and $\mathrm{ct}_{\bar{B}}$, respectively. The ciphertexts $\mathrm{ct}_Z$, $\mathrm{ct}_\beta^{(0)}$ and $\mathrm{ct}_{\bar{B}}$ are as follows:

$$
\mathrm{X} = \begin{bmatrix} 1 & x_{11} & \cdots & x_{1d} \\ 1 & x_{21} & \cdots & x_{2d} \\ \vdots & \vdots & \ddots & \vdots \\ 1 & x_{n1} & \cdots & x_{nd} \end{bmatrix}, \mathrm{Y} = \begin{bmatrix} y_1 \\ y_2 \\ \vdots \\ y_n \end{bmatrix}, \qquad \mathrm{ct}_Z = Enc \begin{bmatrix} y_1 & y_1 x_{11} & \cdots & y_1 x_{1d} \\ y_2 & y_2 x_{21} & \cdots & y_2 x_{2d} \\ \vdots & \vdots & \ddots & \vdots \\ y_n & y_n x_{n1} & \cdots & y_n x_{nd} \end{bmatrix},
$$

$$
\mathrm{ct}_\beta^{(0)} = Enc \begin{bmatrix} \beta_0^{(0)} & \beta_1^{(0)} & \cdots & \beta_d^{(0)} \\ \beta_0^{(0)} & \beta_1^{(0)} & \cdots & \beta_d^{(0)} \\ \vdots & \vdots & \ddots & \vdots \\ \beta_0^{(0)} & \beta_1^{(0)} & \cdots & \beta_d^{(0)} \end{bmatrix}, \qquad \mathrm{ct}_{\bar{B}} = Enc \begin{bmatrix} \bar{B}_{[0][0]} & \bar{B}_{[1][1]} & \cdots & \bar{B}_{[d][d]} \\ \bar{B}_{[0][0]} & \bar{B}_{[1][1]} & \cdots & \bar{B}_{[d][d]} \\ \vdots & \vdots & \ddots & \vdots \\ \bar{B}_{[0][0]} & \bar{B}_{[1][1]} & \cdots & \bar{B}_{[d][d]} \end{bmatrix},
$$

where $\bar{B}_{[i][i]}$ is the diagonal element of $\bar{B}$ that is built from $-\frac{1}{4} X^\top X$.

The pulbic cloud takes the three ciphertexts $\mathrm{ct}_Z$, $\mathrm{ct}_\beta^{(0)}$ and $\mathrm{ct}_{\bar{B}}$ and evaluates the enhanced NAG algorithm to find a decent weight vector by updating the vector $\mathrm{ct}_\beta^{(0)}$. Refer to (Kim et al., 2018a) for a detailed description about how to calculate the gradient by HE programming.

**Limitations** In a privacy-preserving setting, when compared to the NAG method, the primary limitation of the Enhanced NAG method is that it requires one additional ciphertext multiplication to construct the quadratic gradient. In addition, the data owner needs to upload one more ciphertext encrypting the matrix $\bar{B}$. However, the enhanced algorithm converges faster, and we believe it can compensate for the mentioned shortcomings.

Table 1: Implementation Results for iDASH datasets with 10-fold CV

| Dataset | Sample Num | Feature Num | Method | Iter Num | Storage (GB) | Learn Time (min) | Accuracy (%) | AUC |
|---------|-----------|-------------|--------|----------|--------------|------------------|--------------|-----|
| iDASH | 1579 | 18 | Ours | 4 | 0.08 | 4.43 | 61.46 | 0.696 |
| | | | Baseline | 7 | 0.04 | 6.07 | 62.87 | 0.689 |

Table 2: Implementation Results for other datasets with 5-fold CV

| Dataset | Sample Num | Feature Num | Method | Iter Num | Storage (GB) | Learn Time (min) | Accuracy (%) | AUC |
|---------|-----------|-------------|--------|----------|--------------|------------------|--------------|-----|
| Edin | 1253 | 8 | Ours | 4 | 0.04 | 0.6 | 89.52 | 0.943 |
| | | | Baseline | 7 | 0.02 | 3.6 | 91.04 | 0.958 |
| lbw | 189 | 8 | Ours | 4 | 0.04 | 0.6 | 71.35 | 0.667 |
| | | | Baseline | 7 | 0.02 | 3.3 | 69.19 | 0.689 |
| nhanes3 | 15649 | 15 | Ours | 4 | 0.31 | 4.5 | 79.23 | 0.637 |
| | | | Baseline | 7 | 0.16 | 7.3 | 79.22 | 0.717 |
| pcs | 379 | 9 | Ours | 4 | 0.04 | 0.6 | 63.20 | 0.733 |
| | | | Baseline | 7 | 0.02 | 3.5 | 68.27 | 0.740 |
| uis | 575 | 8 | Ours | 4 | 0.04 | 0.6 | 74.43 | 0.597 |
| | | | Baseline | 7 | 0.02 | 3.5 | 74.44 | 0.603 |

## 5 EXPERIMENTS

**Implementation**   We implement the enhanced NAG based on HE with the library `HEAAN`. The C++ source code is publicly available at https://anonymous.4open.science/r/IDASH2017-245B . All the experiments on the ciphertexts were conducted on a public cloud with 32 vCPUs and 64 GB RAM.

For a fair comparison with the baseline (Kim et al., 2018a), we utilized the same 10-fold cross-validation (CV) technique on the same iDASH dataset consisting of 1579 samples with 18 features and the same 5-fold CV technique on the other five datasets. Like (Kim et al., 2018a), We consider the average accuracy and the Area Under the Curve (AUC) as the main indicators. Tables 1 and 2 display the results of the two experiments, respectively. The two tables also provide the average evaluation running time for each iteration and the storage (encrypted dataset for the baseline work and encrypted dataset and $\bar{B}$ for our method). We adopt the same packing method that Kim et al. (2018a) proposed and hence our solution has similar storage of ciphertexts to (Kim et al., 2018a) with some extra ciphertexts to encrypt the $\bar{B}$. We chose $1 + 0.9^t$ as our learning rate configuration.

The parameters of `HEAAN` we set are same to (Kim et al., 2018a): $logN = 16$, $logQ = 1200$, $logp = 30$, $slots = 32768$, which ensure the security level $\lambda = 80$. We use a larger $logp = 40$ to encrypt the matrix $\bar{B}$ for preserving the precision of $\bar{B}$. Refer to (Kim et al., 2018a) for details on these parameters. Since our enhanced NAG method need one more ciphertext multiplication than the baseline work, consuming more modulus, our solution thus can only perform $4$ iterations of the enhanced NAG method. Yet despite only $4$ iterations, our enhanced NAG method still produces a comparable result.

## 6 CONCLUSION

In this paper, we proposed a faster gradient variant, termed `quadratic gradient`, and implemented the quadratic-gradient version of NAG in the encrypted domain to train a logistic regression model. The quadratic gradient introduced in this work can be directly constructed from the Hessian, effectively merging first-order gradient (descent) methods with the second-order Newton's method. There is a promising potential for the quadratic gradient to accelerate other gradient methods, and it may serve as an alternative or even a replacement for the traditional line-search method.

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
