# OpenReview forum: "Privacy-Preserving Logistic Regression Training with A Faster Gradient Variant"
_ICLR.cc/2025/Conference — ICLR 2025 Conference Withdrawn Submission_

### Official Review · Reviewer_fG5a · 2024-10-30

**Soundness:** 3
**Presentation:** 3
**Contribution:** 3
**Rating:** 5
**Confidence:** 2

**Summary:**

This paper introduces an enhanced NAG (Nesterov Accelerated Gradient) method for efficient training of logistic regression, specifically designed for secure training under homomorphic encryption. The empirical study shows that the proposed method achieves comparable results in just four iterations.

**Strengths:**

- S1: The proposed enhanced NAG method is well-analyzed theoretically.
- S2: The advantages of the enhanced NAG method are demonstrated empirically in both plaintext and ciphertext settings. Under homomorphic encryption, it achieves comparable accuracy to baselines with improved efficiency in learning time and iteration count.
- S3: The paper is well-organized and easy to follow.

**Weaknesses:**

- W1: The related work section lacks sufficient detail. In particular, the relationship to other learning algorithms under FHE, beyond just NAG, should be more clearly described.
- W2: The experimental baseline is limited to a single method. Given the existence of various homomorphic computation methods, comparisons with these alternatives are necessary to establish the novelty of the proposed approach.
- W3: Although the paper shows that the proposed method accelerates learning computation in FHE, the improvement in performance is modest, and there is a slight drop in accuracy. The paper should also discuss potential applications where these specific efficiency gains are essential.

**Questions:**

Address W1, W2, and W3

**Details Of Ethics Concerns:**

Nothing

---

> ### Author Response · Authors · 2024-11-24
>
> We sincerely thank the reviewers for their thoughtful and constructive feedback.
>
> ---
>
> #### **Comment 1:**
> *The related work section lacks sufficient detail. In particular, the relationship to other learning algorithms under FHE, beyond just NAG, should be more clearly described.*
>
> **Response:**
> The first version of our paper was nearly finished in 2019, at a time when there were relatively few machine learning algorithms developed for FHE. Also, we wrote the initial version in a rush and did not thoroughly detail other relevant works that were already included in the literature at the time.
>
> We acknowledge that our paper lacks a sufficient review of recent developments in this field. We should have conducted a more comprehensive survey of recent machine learning algorithms that use FHE, as these would provide important context for our work.
>
> ---
>
> #### **Comment 2:**
> *The experimental baseline is limited to a single method. Given the existence of various homomorphic computation methods, comparisons with these alternatives are necessary to establish the novelty of the proposed approach.*
>
> **Response:**
> There are indeed various homomorphic computation methods, and a comparison with these alternatives would strengthen the presentation of our approach’s novelty. However, we believe that, at present, there are not many advanced machine learning algorithms for classification that can be efficiently trained in an FHE setting. Early homomorphic encryption algorithms, such as those focused on computing statistical parameters, differ significantly from the methods explored in both the baseline work and our own research. As such, a direct comparison with these methods might not be entirely appropriate.
>
> However, we agree that it would be beneficial to briefly mention and provide context for some of the existing work in the related work section. This will help to clarify the landscape of homomorphic encryption in machine learning.
>
> ---
>
> #### **Comment 3:**
> *Although the paper shows that the proposed method accelerates learning computation in FHE, the improvement in performance is modest, and there is a slight drop in accuracy. The paper should also discuss potential applications where these specific efficiency gains are essential.*
>
> **Response:**
> Accuracy may not fully reflect the convergence speed of an algorithm. In our case, the proposed enhanced NAG method incorporates additional computation due to the quadratic gradient, which results in fewer iterations compared to the baseline method. This, in turn, may lead to a slight reduction in accuracy. However, we believe that the trade-off is justified by the faster convergence and reduced computational time, which are particularly important in privacy-preserving settings under homomorphic encryption.
>
> Furthermore, the baseline method performs more iterations, which allows it to achieve a higher accuracy at the cost of longer computation. This discrepancy in the number of iterations contributes to the modest difference in performance between the two algorithms.
>
> Finally, the randomization inherent in cross-validation introduces some variability in the results, making it challenging to establish a decisive performance difference between the two methods.
>
> ---
>
> We once again thank the reviewers for their thoughtful feedback. These comments have been instrumental in improving our work, and we are confident the revised manuscript will address your concerns comprehensively.

---

### Official Review · Reviewer_DNPX · 2024-10-30

**Soundness:** 2
**Presentation:** 2
**Contribution:** 1
**Rating:** 3
**Confidence:** 4

**Summary:**

This paper proposes a preconditioning scheme for accelerating gradient-descent style algorithms. The paper explains how to incorporate this scheme into the logistic regression fitting with homomorphic encryption framework introduced in prior work (Kim et al.  2018a).

Experiments on a variety of datasets in very tiny plots show that the proposed preconditioner improves the training convergence and/or training loss for nesterov accelerated gradient, adagrad, and adam. However, comparisons in the homomorphic encryption framework to Kim et al. are inconclusive as often the Kim et al. version can improve accuracy at a cost of a few more minutes of training.

**Strengths:**

This paper proposes a preconditioner that improves the training loss or training convergence of 3 different styles of gradient descent on 8 datasets.

**Weaknesses:**

Before discussing  weaknesses, it is important to add some context to the paper. Multiplying the gradient by a positive (or negative) definite matrix during gradient descent/ascent is known as preconditioning.  This is perhaps the connection the paper was looking for on Line 178. Thus this paper is mostly about finding a satisfactory preconditioner. With this context, one can observe the following weaknesses.

1) The literature on preconditioning is huge and classical texts such as Numerical Optimization by Nocedal and Wright discuss it in depth. When the main contribution is a preconditioner, not only does the author need to start with the classical practical preconditioners mentioned there, but also conduct a literature search to identify what's new and check if this specific preconditioner has already been proposed. On top of that, since so many preconditioners and variations are available, it is always possible to find something that works well for any handful of datasets and a theoretical analysis of when it is superior is needed.

2) The proposed preconditioner is a small variation of Kim et al. As far as I can tell, it adds absolute values and an additive epsilon term that is a fairly common techniques for making matrices more positive definite (or negative definite, depending on the sign).

3) The paper does not propose anything new on the privacy side. The framework of Kim et al. is re-used.

4) The paper offloads some computation on the client, specifically the computation of the preconditioner. It raises the question of what is reasonable to offload on the client and what is not.
- One possibility is to offload nothing onto the client -- the client simply uploads the dataset to the cloud for long term storage and the cloud servers can provide various analytics functionality in the future. However, for this setting the paper needs a privacy preserving preconditioner computation.
- On the other hand, if the client is already computing the preconditioner, why wouldn't the client run gradient descent (or sdg for large datasets) to train the logistic regression themselves? There are many software packages that do this and it is a very fast computation.

5) Most of the experiments in the paper suspiciously only focus on the training loss and the paper explicitly refuses to examine other metrics, such as training classification error, testing loss, and testing classification error. Since loss and classification error are not perfectly correlated, and can exhibit strange behaviors when an optimizer is changed, it raises red flags that the paper should avoid (by running these more complete experiments).

**Questions:**

I do not have any questions.

---

> ### Author Response · Authors · 2024-11-24
>
> We sincerely thank the reviewers for their thoughtful and constructive feedback.
>
> ---
>
> #### **Comment 1:**
> *The paper offloads some computation on the client, specifically the computation of the preconditioner. It raises the question of what is reasonable to offload on the client and what is not.
> One possibility is to offload nothing onto the client -- the client simply uploads the dataset to the cloud for long term storage and the cloud servers can provide various analytics functionality in the future. However, for this setting the paper needs a privacy preserving preconditioner computation.
> On the other hand, if the client is already computing the preconditioner, why wouldn't the client run gradient descent (or sdg for large datasets) to train the logistic regression themselves? There are many software packages that do this and it is a very fast computation.*
>
> **Response:**
> In machine learning applications, it is common practice to preprocess the dataset before applying any algorithms. Simply encrypting and uploading a raw dataset to the cloud for processing is not recommended.
>
> In a reasonable scenario, basic data preprocessing, including data cleaning and the calculation of $\bar B$ , could be provided as a free service. More advanced operations, such as logistic regression training, could then be offered as a premium, chargeable service.
>
> ---
>
> #### **Comment 2:**
> *Most of the experiments in the paper suspiciously only focus on the training loss and the paper explicitly refuses to examine other metrics, such as training classification error, testing loss, and testing classification error. Since loss and classification error are not perfectly correlated, and can exhibit strange behaviors when an optimizer is changed, it raises red flags that the paper should avoid (by running these more complete experiments).*
>
> **Response:**
> We focus on the convergence speed of our gradient variant and aim to study its direct impact on the loss function. The outputs of the loss function provide a perfect measure for this purpose. In contrast, metrics such as training classification error, testing loss, and testing classification error do not offer the same insights.
>
> For instance, in some cases, the training classification error may remain unchanged even as the loss function increases or decreases. Similarly, testing loss and testing classification error are unrelated to the performance of the loss function on the training data and are more relevant from the perspective of machine learning applications. Overfitting provides a clear example of this disconnect: the testing loss may behave inconsistently with the training loss, highlighting their distinct roles.
>
> ---
>
> We guess that a theoretical analysis of our enhanced methods is related to the fixed Hessian method only, rather than preconditioning. The fundamental tool for analyzing the fixed Hessian method is a geometric insight, obtained by interpreting quadratic-approximation algorithms as a form of area approximation rather than using a purely mathematical approach. This poses a significant challenge to the analysis of our enhanced algorithms. Since the proposed gradient is based on the fixed Hessian method, other optimization theories might not provide assistance.
>
>
>
> ---
>
> We once again thank the reviewers for their thoughtful feedback. These comments have been instrumental in improving our work, and we are confident the revised manuscript will address your concerns comprehensively.

---

### Official Review · Reviewer_HLXf · 2024-11-03

**Soundness:** 3
**Presentation:** 2
**Contribution:** 2
**Rating:** 3
**Confidence:** 4

**Summary:**

This paper presents a secure training algorithm for logistic regression models. Specifically, the proposed training algorithm uses a gradient estimator called quadratic gradient which preconditions the gradient using the estimated second-order information. To reduce the computational burden, the second-order information (i.e., Hessian) is approximated as a diagonal matrix. The training algorithm is implemented with a homomorphic encryption framework.

**Strengths:**

- The proposed algorithm employs the fixed diagonal approximation of Hessian which reduces computational cost for calculating the Hessian during the training and can be approximated in the encrypted form.
- The proposed quadratic gradient is applied to three optimization algorithms: Nesterov’s Accelerated Gradient (NAG), AdaGrad, and Adam, and their performance is compared across various datasets.

**Weaknesses:**

- The proposed quadratic gradient seems to be similar to the preconditioned gradient, which has long been utilized in the optimization community, and there exists a substantial body of literature on this topic. The proposed work seems to make an incremental contribution, as the quadratic gradient appears to be a rebranding of the existing preconditioned gradient. In addition, the idea of approximating the parameter-dependent Hessian matrix with a parameter-independent input covariance matrix was introduced in the literature. For example,
  1. Ida, Yasutoshi, Yasuhiro Fujiwara, and Sotetsu Iwamura. “Adaptive Learning Rate via Covariance Matrix Based Preconditioning for Deep Neural Networks.” In Proceedings of the Twenty-Sixth International Joint Conference on Artificial Intelligence, IJCAI-17
  2. Mehta, Harsh, Walid Krichene, Abhradeep Guha Thakurta, Alexey Kurakin, and Ashok Cutkosky. “Differentially Private Image Classificafrom Features.” Transactions on Machine Learning Research, November 29, 2022.

- The proposed algorithm is implemented using the existing homomorphic encryption framework. It is unclear what the significance of the use of quadratic gradient in the context of homomorphic encryption.
- The empirical results are not well explained. The performance comparison with the existing algorithm by Kim et al. shows that the proposed algorithm achieves lower accuracy than the baseline method on some datasets (for example, Edin and pcs datasets).
- The presentation of the paper can be improved.

**Questions:**

- The empirical results presented in Section 5 require further explanation. The proposed method uses twice as much memory as the baseline, which is understandable given that it needs to store diagonal entries of the preconditioning matrix. However, it seems to be faster than the baseline in terms of the learn time. What exactly does the learn time measure and why is it the case? Given that the proposed algorithm includes an addition step compared to the baseline, it is not clear why the proposed algorithm is faster.
- What is the motivation for approximating the diagonal entries of Hessian with the row-wise sum of $\bar{H}$?
- I am not sure how the formulation presented in line 176 - 181 help better explain the idea of proposed work.

---

> ### Author Response · Authors · 2024-11-24
>
> We sincerely thank the reviewers for their thoughtful and constructive feedback.
>
> ---
>
> #### **Comment 1:**
> *The empirical results presented in Section 5 require further explanation. The proposed method uses twice as much memory as the baseline, which is understandable given that it needs to store diagonal entries of the preconditioning matrix. However, it seems to be faster than the baseline in terms of the learn time. What exactly does the learn time measure and why is it the case? Given that the proposed algorithm includes an addition step compared to the baseline, it is not clear why the proposed algorithm is faster.*
>
> **Response:**
> The primary reason our method required less learning time is that it was run for only 4 iterations, compared to the baseline work's 7 iterations. Additionally, another potential factor is the difference in computational resources: we ran our algorithms on a server with 32 vCPUs, whereas the baseline work used an Intel Xeon CPU E5-2620 v4 at 2.10 GHz (likely with 16 vCPUs). From their open-source code, it appears they utilized 8 threads for computation.
>
> We apologize for the oversight in not thoroughly analyzing the baseline's performance, as we did not focus heavily on comparing learning times. Our learning time measurement reflects the time required for 4 iterations of our enhanced algorithms. Upon further review, we noticed that the baseline's reported average running time includes time for evaluation and encryption. However, it remains unclear whether their evaluation time accounts for the total computation across all folds or just the 7 iterations required for a single fold.
>
> It is worth noting that we relied on their open-source code to implement our homomorphic logistic regression training. Therefore, theoretically, the running time for one iteration of our algorithm should be very similar to theirs.
>
> ---
>
> #### **Comment 2:**
> *What is the motivation for approximating the diagonal entries of Hessian with the row-wise sum of $\bar H$?*
>
> **Response:**
> The motivation for approximating the diagonal entries of the Hessian matrix with the row-wise sum of $\bar H$ is to construct a diagonal matrix that satisfies the conditions of the (Simplified) Fixed Hessian method. Our gradient variants and methods are fundamentally based on the (Simplified) Fixed Hessian framework.
>
> Specifically, our enhanced NAG method can be interpreted as a Simplified Fixed Hessian method with an aggressive learning rate, which drives its faster convergence under suitable conditions.
>
> ---
>
> #### **Comment 2:**
> *I am not sure how the formulation presented in line 176 - 181 help better explain the idea of proposed work.*
>
> **Response:**
> The formulation presented in Lines 176–181 initially helped the authors recognize the connection between Newton's method and Standard Gradient Descent. This insight inspired the foundational idea of this work: converting the Hessian into a gradient-like form.
>
> Although this formulation may appear less directly related to the later ideas in the paper, we chose to include it in the manuscript as it may similarly inspire other readers in their research.
>
>
>
> ---
>
> We once again thank the reviewers for their thoughtful feedback. These comments have been instrumental in improving our work, and we are confident the revised manuscript will address your concerns comprehensively.

---

### Official Review · Reviewer_JbEc · 2024-11-04

**Soundness:** 2
**Presentation:** 3
**Contribution:** 2
**Rating:** 3
**Confidence:** 3

**Summary:**

The paper proposes a new optimization technique for privacy-preserving logistic regression training called quadratic gradient. This method combines elements from first order gradient algorithms and the second-order Newton-Raphson method. The variant constructs a diagonal Hessian approximation that allows for the computation of an optimized gradient update without recalculating the computationally expensive Hessian matrix. As a result, the quadratic gradient facilitates faster convergence by leveraging curvature information while retaining the simplicity of first-order methods.
The authors show that quadratic-gradient versions of Nesterov Accelerated Gradient (NAG), Adagrad, and Adam optimizers converge, and, in most cases, demonstrate strong performance in terms of convergence speed, by empirical evaluation on multiple datasets. The authors apply the quadratic gradient-enhanced NAG to homomorphic encryption-based logistic regression training, showing comparable model accuracy within only a few iterations.

**Strengths:**

The quadratic gradient approach provides a new combination of first-order gradient and second-order Newton-Raphson methods. Multiple experiments were considered to demonstrate the improvements in convergence speed achieved by the enhanced NAG, Adagrad, and Adam algorithms. The paper aims to address a practical challenge in encrypted logistic regression training, where efficient computation is crucial.
The quadratic gradient concept builds on several ideas from the literature, which are well reported in the paper.

**Weaknesses:**

- The claim that the enhanced algorithms achieve significantly improved convergence speed compared to traditional first-order gradient
methods is not very well supported, as the performance of the algorithms varies across different datasets. The enhanced Adagrad and Adam methods do not achieve faster convergence on the iDASH, and while they do perform better on other datasets, it is hard to assess the overall extent and consistency of the improvement. Namely, the authors mention: _"Although the enhanced Adagrad and Adam methods do not achieve faster convergence than their original counterparts on the iDASH genomic dataset, they demonstrate clear advantages in all other cases. The reason for this discrepancy is beyond the scope of this paper and will be addressed as part of future work."_. It is worth providing additional analysis in this work on why certain datasets showed less improvement, as it would help identify conditions under which the method is most effective (i.e., how do the methods behave under different number of dimensions, dataset size, etc.). Providing bounds on convergence speed or computational efficiency would also strengthen the contribution.
- The paper's writing makes its focus somewhat unclear. The quadratic gradient method is proposed specifically as a technique for privacy-preserving logistic regression, but a significant portion of the paper is dedicated to showing that quadratic gradient can perform better than first-order gradient methods in the general (non-private) setting. The private setting is proposed as an application.

**Questions:**

- Further insights (through experiments and/or theory) should be provided on the performance of the quadratic gradient approach (see weaknesses).

---

> ### Author Response · Authors · 2024-11-24
>
> We sincerely thank the reviewers for their thoughtful and constructive feedback.
>
> ---
>
> #### **Comment 1:**
> *The claim that the enhanced algorithms achieve significantly improved convergence speed compared to traditional first-order gradient methods is not very well supported, as the performance of the algorithms varies across different datasets. The enhanced Adagrad and Adam methods do not achieve faster convergence on the iDASH, and while they do perform better on other datasets, it is hard to assess the overall extent and consistency of the improvement. Namely, the authors mention: "Although the enhanced Adagrad and Adam methods do not achieve faster convergence than their original counterparts on the iDASH genomic dataset, they demonstrate clear advantages in all other cases. The reason for this discrepancy is beyond the scope of this paper and will be addressed as part of future work.". It is worth providing additional analysis in this work on why certain datasets showed less improvement, as it would help identify conditions under which the method is most effective (i.e., how do the methods behave under different number of dimensions, dataset size, etc.). Providing bounds on convergence speed or computational efficiency would also strengthen the contribution.*
>
> **Response:**
> Providing additional analysis on why certain datasets showed less improvement for the enhanced Adagrad and Adam methods is currently beyond the authors' capability. Despite our efforts, we were unable to identify the specific conditions under which these two enhanced methods are most effective. This limitation is a shortcoming of our work, and we acknowledge that it leaves the study incomplete in this regard.
>
> We hope that future research will address this issue and provide deeper insights. Another fundamental challenge is proving the convergence of the enhanced Adagrad and Adam algorithms, which remains an unresolved concern for us as well.
>
> ---
>
> #### **Comment 2:**
> *The paper's writing makes its focus somewhat unclear. The quadratic gradient method is proposed specifically as a technique for privacy-preserving logistic regression, but a significant portion of the paper is dedicated to showing that quadratic gradient can perform better than first-order gradient methods in the general (non-private) setting. The private setting is proposed as an application.*
>
> **Response:**
> We focused primarily on the quadratic gradient methods rather than their application in the encrypted setting. Preliminary experiments have demonstrated that our enhanced methods can also be applied to numerical optimization problems beyond privacy-preserving scenarios.
>
> When using homomorphic encryption, running algorithms in the private setting is largely similar to the non-private setting. For instance, we first analyzed our algorithms in the clear using Python, replacing the sigmoid function with its polynomial approximation. After determining the appropriate parameters, we implemented privacy-preserving logistic regression training in the encrypted domain using C++ and the HEAAN library.
>
> The baseline work introduced several excellent homomorphic encryption techniques, and we utilized all the optimizations they proposed. We believe that no additional techniques are required for homomorphic logistic regression training beyond those already developed.
>
>
> ---
>
> We once again thank the reviewers for their thoughtful feedback. These comments have been instrumental in improving our work, and we are confident the revised manuscript will address your concerns comprehensively.

---

### Official Review · Reviewer_QAC5 · 2024-11-04

**Soundness:** 2
**Presentation:** 3
**Contribution:** 2
**Rating:** 3
**Confidence:** 3

**Summary:**

The authors propose a privacy-preserving approach to training logistic regression based on homomorphic encryption. Their approach utilises second-order information through quadratic gradients to implement various optimisers such as NAG/Adagrad and claim this helps achieve improved convergence speeds compared to first-order methods.

**Strengths:**

- The paper tackles an important and useful problem, focusing on improving the convergence speed of privacy-preserving logistic regression.
- The paper is clearly written and well-structured.

**Weaknesses:**

- The evaluation in the privacy-preserving setting is limited. Most of the experiments focus on the non-private setting and the private experiments compare only to a single baseline method.
- The empirical results are often unconvincing. It is not clear that the proposed quadratic gradient approach gives any clear benefit on some of the benchmark datasets in terms of convergence speed or accuracy/AUC.
- There is not enough of a comparison with prior privacy-preserving logistic regression methods to enable a clear understanding of why this method should be used compared to prior work.

**Questions:**

1. In Tables 1 and 2, the proposed enhanced NAG method generally performs worse than the baseline method for accuracy and AUC. It is also mentioned then when compared to the Kim et al. baseline, the enhanced NAG requires one additional ciphertext multiplication and so you run it for less iterations.  Is the only advantage of the enhanced NAG computation time?
2. In addition to the point above, there could be a more thorough set of experiments for the privacy-preserving setting. Restricting the comparison to 4 iterations vs. 7 does not do enough to highlight when the enhanced NAG method should be used. There is a likely trade-off in terms of convergence and compute that is not being fully explored in the experiments.
3. As far as I can tell, the proposed quadratic gradient method is a straightforward extension to the work of Bonte and Vercauteren (2018). Why do you not compare to their method in Section 5? I feel the privacy-preserving experiments are lacking further baselines.
4. In Line 185, is the gradient meant to have a subscript i here? Is is meant to be referring to $\nabla_\beta$?
5. Minor point: There are a few typos that could be corrected — L477: pulbic → public, L248: practicable -> practical, L269: setted → set

---

> ### Author Response · Authors · 2024-11-24
>
> We sincerely thank the reviewers for their thoughtful and constructive feedback.
>
> ---
>
> #### **Comment 1:**
> *In Tables 1 and 2, the proposed enhanced NAG method generally performs worse than the baseline method for accuracy and AUC. It is also mentioned that when compared to the Kim et al. baseline, the enhanced NAG requires one additional ciphertext multiplication and so you run it for fewer iterations. Is the only advantage of the enhanced NAG computation time?*
>
> **Response:**
> The primary advantage of the enhanced NAG lies in its faster convergence speed. Compared to the raw NAG method, the enhanced NAG achieves comparable performance within fewer iterations, significantly reducing computation time and eliminating the need for additional iterations.
>
> However, the enhanced NAG introduces one additional ciphertext multiplication to compute the quadratic gradient, which increases modulus consumption. Since the baseline work did not incorporate ciphertext refreshing techniques, we also chose not to apply bootstrapping. As a result, we limited the number of iterations in our experiments, which led to slightly worse performance than the baseline method in terms of accuracy and AUC, as shown in Tables 1 and 2.
>
> ---
>
> #### **Comment 2:**
> *In addition to the point above, there could be a more thorough set of experiments for the privacy-preserving setting. Restricting the comparison to 4 iterations vs. 7 does not do enough to highlight when the enhanced NAG method should be used. There is a likely trade-off in terms of convergence and compute that is not being fully explored in the experiments.*
>
> **Response:**
> We assume that both 4 and 7 iterations are sufficient for demonstrating the method’s effectiveness in our experiments. In real-world applications, however, the bootstrapping technique is typically employed to enable sufficient iterations. The baseline work was designed as a solution for the iDASH competition, which did not support ciphertext refreshing.
>
> In practice, we prefer to run logistic regression algorithms—whether enhanced or not—for as many iterations as possible, typically at least several dozen, to ensure a well-trained and reliable model.
>
> ---
>
> #### **Comment 3:**
> *As far as I can tell, the proposed quadratic gradient method is a straightforward extension to the work of Bonte and Vercauteren (2018). Why do you not compare to their method in Section 5? I feel the privacy-preserving experiments are lacking further baselines.*
>
> **Response:**
> The work of Bonte and Vercauteren (2018), namely the Simplified Fixed Hessian (SFH) method, cannot be applied to certain datasets, such as MNIST. Our proposed quadratic gradient method builds upon their SFH approach by introducing a new learning rate greater than 1. Predictably, a larger learning rate often results in better performance compared to the SFH method, provided convergence is achieved.
>
> The comparison to their method is straightforward, and the outcome is predictable. In fact, the simplest implementation of our enhanced gradient method is nearly identical to the SFH method. However, we acknowledge that conducting a formal comparison would strengthen our work, especially as it highlights the limitations of SFH on datasets like MNIST.
>
> ---
>
> #### **Comment 4:**
> *In Line 185, is the gradient meant to have a subscript $ i$ here? Is it meant to be referring to $\nabla_{\beta}$?*
>
> **Response:**
> We apologize for this oversight. No, the gradient in Line 185 is not intended to have a subscript $ i$ there . It is meant to refer to $\nabla_{\beta}$.
>
>
>
> ---
>
> #### **Comment 5:**
> *Minor point: There are a few typos that could be corrected — L477: pulbic → public, L248: practicable → practical, L269: setted → set.*
>
> **Response:**
> We appreciate your attention to these errors. All identified typos will be corrected in the next submission.
>
> ---
>
> We once again thank the reviewers for their thoughtful feedback.

---

### Note · Authors · 2024-11-24

**Comment:**

I am writing to formally request the withdrawal of our submitted paper, titled *"Privacy-Preserving Logistic Regression Training with A Faster Gradient Variant"* with ID 2608, from the ICLR 2025 Conference.

After careful consideration, we have identified certain limitations in our work that we believe need to be addressed to ensure a higher standard of contribution. Specifically, the paper lacks a comprehensive review of recent related works and a broader experimental comparison with other homomorphic encryption methods, which are essential for a robust evaluation of our approach. We feel it is necessary to refine and improve the work further before submitting it for peer review.

We sincerely apologize for any inconvenience caused and appreciate your understanding. Please confirm if there are any additional steps we need to follow.

Best regards,
On behalf of all co-authors

**Withdrawal Confirmation:**

I have read and agree with the venue's withdrawal policy on behalf of myself and my co-authors.